# Frequency of non-communicable diseases in people 50 years of age and older receiving HIV care in Latin America

Pablo F. Belaunzaran-Zamudio[1], Yanink Caro-Vega[1]*, Mark J. Giganti[2], Jessica L. Castilho[3], Brenda E. Crabtree-Ramirez[1], Bryan E. Shepherd[2], Fernando Mejía[4], Carina Cesar[5], Rodrigo C. Moreira[6], Marcelo Wolff[7], Jean W. Pape[8], Denis Padgett[9], Catherine C. McGowan[3], Juan G. Sierra-Madero[1], for the Caribbean, Central and South American network for HIV epidemiology (CCASAnet)¶

1 Departamento de Infectología, Instituto Nacional de Ciencias Médicas y Nutrición Salvador Zubirán, Mexico City, Mexico, 2 Department of Biostatistics, Vanderbilt University, Nashville, Tennessee, United States of America, 3 Department of Medicine, Vanderbilt University, Nashville, Tennessee, United States of America, 4 Instituto de Medicina Tropical Alexander von Humboldt, Universidad Peruana Cayetano Heredia, Lima, Peru, 5 Fundación Huesped, Buenos Aires, Argentina, 6 Fundação Oswaldo Cruz, Instituto Nacional de Infectologia Evandro Chagas, Rio de Janeiro, Brazil, 7 Department of Infectious Diseases, Fundación Arriarán, Santiago de Chile, Chile, 8 Les Centres GHESKIO, Port-au-Prince, Haiti, 9 Instituto Hondureño de Seguridad Social, Tegucigalpa, Honduras

¶ Membership of the Caribbean, Central and South American network for HIV epidemiology (CCASAnet) is provided in the Acknowledgments.

* yanink.caro@infecto.mx

**Data Availability Statement:** Complete data for this study cannot be publicly shared because of

## Abstract

### Background

A growing population of older adults with HIV will increase demands on HIV-related health-care. Nearly a quarter of people receiving care for HIV in Latin America are currently 50 years or older, yet little is known about the frequency of comorbidities in this population. We estimated the prevalence and incidence of non-communicable diseases (NCDs) among people 50 years of age or older (≥50yo) receiving HIV care during 2000–2015 in six centers affiliated with the Caribbean, Central and South American network for HIV epidemiology (CCASAnet).

### Methods

We estimated the annual prevalence, and overall prevalence and incidence of cardiovascular diseases, diabetes, hypertension, dyslipidemia, psychiatric disorders, chronic liver and renal diseases, and non-AIDS-defining cancers, and multimorbidity (more than one NCD) of people ≥50yo receiving care for HIV. Analyses were performed according to age at enrollment into HIV care (<50yo and ≥50yo).

### Results

We included 3,415 patients ≥50yo, of whom 1,487(43%) were enrolled at age ≥50 years. The annual prevalence of NCDs increased from 32% to 68% and multimorbidity from 30%

legal and ethical restrictions. The Principles of Collaboration under which the CCASAnet multi-national collaboration was founded and the regulatory requirements of the different countries' IRBs require the submission and approval of a project concept sheet by the CCASAnet Executive Committee and the principal investigators at participating sites. All datasets provided by CCASAnet are de-identified according to HIPAA Safe Harbor guidelines. Since re-identification of de-identified datasets may be possible when they are combined with publicly available datasets, CCASAnet promotes the signing of a Data Use Agreement before HIV clinical data can be released. Instructions for how to obtain CCASAnet data are outlined on the CCASAnet website: https://www.ccasanet.org/collaborate/.

**Funding:** This work was supported by the National Institutes of Health–funded Caribbean, Central and South America network for HIV epidemiology (CCASAnet), a member cohort of the International Epidemiology Databases to Evaluate AIDS (IeDEA; U01AI069923). This award is funded by the following institutes: Eunice Kennedy Shriver National Institute of Child Health and Human Development (NICHD), Office of the Director (OD), National Institutes of Health, National Institute of Allergy and Infectious Diseases (NIAID), National Cancer Institute (NCI), and the National Institute of Mental Health (NIMH). Dr. Castilho was also supported by K23AI120875. The funders had no role in study design, data collection and analysis, decision to publish, or preparation of the manuscript.

**Competing interests:** Pablo F Belaunzaran-Zamudio, Yanink Caro-Vega, Mark J Giganti, Brenda Crabtree- Ramírez, Bryan E Shepherd, Carina Cesar, Rodrigo C Moreira, Fernando Mejia, Marcelo Wolff, Jean W. Pape, Denis Padgett and Catherine C McGowan have no conflicts to declare. Juan Sierra-Madero reports personal fees and non-financial support from Gilead, non-financial support from MSD, grants from BMS, grants from Pfizer, and personal fees from Jansen, all outside the submitted work. This does not alter our adherence to PLOS ONE policies on sharing data and materials. Complete data for this study cannot be publicly shared because of legal and ethical restrictions. The Principles of Collaboration under which the CCASAnet multi-national collaboration was founded and the regulatory requirements of the different countries' IRBs require the submission and approval of a project concept sheet by the CCASAnet Executive Committee and the principal investigators at participating sites. All datasets provided by CCASAnet are de-identified

to 40% during 2000–2015. At the last registered visit, 53% of patients enrolled <50yo and 50% of those enrolled ≥50yo had at least one NCD. Most common NCDs at the last visit in each age-group at enrollment were dyslipidemia (36% in <50yo and 28% in ≥50yo), hypertension (17% and 18%), psychiatric disorders (15% and 10%), and diabetes (11% and 12%).

## Conclusions

The prevalence of NCDs and multimorbidity in people ≥50 years receiving care for HIV in CCASAnet centers in Latin America increased substantially in the last 15 years. Our results make evident the need of planning for provision of complex, primary care for aging adults living with HIV.

## Introduction

A growing population of older people with HIV is receiving care worldwide [1,2]. People living with HIV (PLWHIV) now have extended life expectances due to use of combined antiretroviral therapies (cART). Meanwhile, the observed age at infection is also increasing [2,3]. By 2030, model-based projections estimate that around 75% of people receiving care for HIV in high-income countries will be older than 50 years [4,5]. As the population of PLWHIV ages, co-morbidities resulting from natural aging, effects of chronic inflammation, and long-term use of cART are expected to also grow, increasing demands on HIV-related health care [5,6].

According to UNAIDS estimates, there were 4.2 million people aged 50 or older living with HIV in low- and middle-income countries (LMIC) by 2013, accounting for 12% of all PLWHIV in these regions [2]; in Latin America and the Caribbean, that proportion may be as high as 24% [7,8]. A recent study by Carraquiry *et al* suggested that those starting cART at 50 years or older had an increased risk of death independent of CD4 count or AIDS when compared with people starting cART at younger ages, despite decreased risk of virological failure and treatment modification [9]. One possible reason for this difference could be the increased presence of non-communicable diseases (NCDs) in the population over 50 years [10].

Among the general population of people aged 50 or older in Latin America, the prevalence of NCDs is high. For example, recent studies from Brazil [11,12], Honduras [13], Mexico [14–16], and Peru [17] have all reported estimates of the prevalence of diabetes mellitus (DM) in older populations to be 12% or higher. Among those between the ages of 60 and 70 years in Mexico, the estimated prevalence was as high as 36% [14]. The percentage of people 50 years and older with hypertension (HTN) was estimated to be greater than 40% in urban and rural settings in Mexico [14–16], Brazil [11,12], and Honduras [13]. Dyslipidemia was also higher than 35% in both Brazil and Mexico [11,12,16].

While the frequency of NCDs among aging PLWHIV has been extensively studied in the USA, Europe, and Africa [4–6, 18–20], few studies have addressed this emerging problem in Latin America [21–25]. Published data from single-center cohorts of people older than 50 receiving care for HIV in Mexico and Brazil found high prevalence of (DM, 20–22%), HTN (33–62%), dyslipidemia in Mexico (57%), and chronic liver (25%) and kidney disease (16%) in Brazil. Moreover, the mean number of comorbidities was 1.4 in Mexico and 2 in Brazil, with an overall prevalence of multimorbidity of 63% in Brazil [22,23]. Our goal is to further characterize the magnitude of this problem in our region using data from five Latin American

according to HIPAA Safe Harbor guidelines. Since reidentification of de-identified datasets may be possible when they are combined with publicly available datasets, CCASAnet promotes the signing of a Data Use Agreement before HIV clinical data can be released. Instructions for how to obtain CCASAnet data are outlined on the CCASAnet website: https://www.ccasanet.org/collaborate/.

countries. The analysis of data from a larger and more heterogeneous population, provides more generalizable results to inform planning care, and estimate costs and needs of HIV-related healthcare in Latin America. Here, we summarize the prevalence of selected NCDs at different time points, and overall incidence among PLWHIV ≥50 years of age receiving HIV care. Patients were stratified by age at enrollment in care (<50 years at enrollment and ≥ 50 years at enrollment) given possible differences in time of ART exposure, previous health history, HIV duration and severity, clinical outcomes, and mortality between people aging in care and people diagnosed at older age [18, 26].

## Methods

### Ethics statement

Institutional review board approval was obtained locally by each participating Caribbean, Central and South American network for HIV Epidemiology (CCASAnet) site and by the CCASAnet Data Coordinating Center at Vanderbilt University. In each of the sites contributing data to this study, ethical regulations and policies permit retrospective analysis of de-identified clinical data without informed consent when research is approved by an Institutional Review Board or appropriately constituted ethics committee. Argentina–Comité de Bioética de Fundación Huésped. Brazil- Comité de Ética em Pesquisa (CEP), Fundacao Oswaldo Cruz, Instituto Nacional de Infectologia Evandro Chagas. Comissao Nacional de Etica em Pesquisa (CONEP). Chile- Comite Etico Científico del Servicio de Salud Metropolitano Central. Haiti-Comité des Droits Humains des Centres GHESKIO, and Weill Cornel Medical College Institutional Review Board.Honduras–Comité de Ética en Investigación Biomédica de la Unidad de Investigación Científica de la Universidad Nacional autónoma de Honduras. México–Comité de Ética en Investigación del Instituto Nacional de Ciencias Médicas y Nutrición Salvador Zubirán. Perú–Comité Institucional de Ética, Universidad Peruana Cayetano Heredia. Vanderbilt–Vanderbilt University Institutional Review Board.

### Cohort settings and procedures

The Caribbean, Central and South American network for HIV Epidemiology (CCASAnet) comprises a consortium of HIV health care centers from seven countries in Latin America (Argentina, Brazil, Chile, Haiti, Honduras, Mexico and Peru) established to share anonymized clinical data with the purpose of studying the epidemiology of HIV outcomes and care in our region [27]. The characteristics of participating centers are widely heterogeneous; and include private, public and non-government organization centers with different funding mechanisms, infrastructure, size, and type of care (primary or tertiary care level), reflecting the different HIV-care models throughout our region [28–32]. Patients enroll, are followed-up, and receive care and treatment in collaborating sites according to local guidelines. Patients are followed from date of cohort enrollment (date of first clinic visit) until date of death or date of last visit (in the case of loss to follow-up).

CCASAnet cohort data is based on medical records used during routine patient care. CCASAnet Standard Operating Procedures are used to consistently collect, code, organize and submit deidentified data to the CCASAnet Data Coordinating Center at Vanderbilt University (VDCC; Nashville, TN, USA) for harmonization and assessment for internal consistency between variables. Non-AIDS defining events are organized in a predefined list that encompasses 14 generic diagnosis: cerebrovascular disease, coronary artery diseases, end-stage renal disorder, cirrhosis or end-stage liver disease, diabetes, non-AIDS-defining neoplasia, osteoporosis or avascular necrosis of bone, Chagas diseases, dengue, malaria, HTLV-1, leishmaniasis, dyslipidemia, and an additional open formatted category to allow for flexibility among all sites.

Non-communicable disease diagnoses are included solely if recorded in the medical record by the provider. Investigators at each site decide in which category any particular event is registered during data collection. Events might have been diagnosed during routine care since the last HIV-care visit, and either registered in the clinical file or self-reported by patients if diagnosed elsewhere. The VDCC performs periodic quality assessment of data collection and validation through data audits to ensure accuracy [27, 33, 34].

## Study population

This was an observational cohort analysis using clinical data from older PLWHIV (≥50 years) enrolled in participating sites in CCASAnet [27]. We included data from patients receiving care at the six centers where data on non-communicable diseases are collected (Fundación Huesped in Argentina, Instituto Nacional de Infectologia Evandro Chagas -Fiocruz in Brazil, Fundación Arriarán in Chile, Instituto Hondureño de Seguridad Social and Hospital Escuela Universitario in Honduras, and Instituto Nacional de Ciencias Médicas y Nutrición Salvador Zubirán in Mexico).

While the classification of 'older PLWHIV" is not explicitly defined, a threshold of 50 years of age is commonly used to indicate an aging or older HIV population [35]. In our own research, we have used this threshold to assess how people enrolled at younger ages and receiving ART have contributed to growth of the older PLWHIV population compared to those enrolled at older ages [7]. We maintain this categorization to explore how much each group might contribute to the prevalence of comorbidities and multi-morbidity in our region. For this study, we included all patients ≥50 years of age in the cohort during the time between 2000 and 2015 who had at least two clinic visits, including one after 50 years of age. Patients were categorized according to their age at clinic enrollment: those <50 years old (yo) at enrollment and retained in care after age 50yo (<50yo at enrollment) and those who enrolled in care at age ≥50 years (≥50yo at enrollment). Observation time was considered the time in care after 50 years of age. The start of study observation time was different for each group: patients ≥50yo at enrollment started follow-up on the registered date of enrollment in care in each center, and patients <50yo at enrollment started follow-up at their first visit after their 50th birthday, regardless of age at diagnosis or date of enrollment in clinical care and ART initiation. All patients were followed-up until the last visit to the center or death.

## Outcomes, definitions and data sources

Information on NCDs for this study was retrieved from the central CCASAnet database. For this analysis, NCDs were assigned to one of eight categories: diabetes (including any registered diagnosis using the term diabetes); dyslipidemia (including diagnoses registered as dyslipidemia, high cholesterol and hyperlipidemia); psychiatric disorders (including diagnoses registered as depression, unspecified depression, depressive episode, panic attack, depressive disorder, schizophrenia, personality disorder); hypertension (including diagnoses registered using the term hypertension); liver diseases (including diagnoses registered as non-AIDS liver event, hepatitis B, hepatitis C, chronic hepatitis C, hepatic steatosis, steatosis, chronic liver disease, liver insufficiency), cardio- and cerebrovascular diseases (any diagnosis registered as cerebrovascular accident or stroke, cardiovascular disease, coronary artery disease, arterial thrombosis, cerebral ischemia, myocardial infarction/thrombotic-ischemic disease, unspecified heart failure, peripheral vascular insufficiency, chest pain, or hypertensive heart disease); renal diseases (all registered non-AIDS related renal disorders, including chronic tubulointerstitial nephritis, diabetic nephropathy, hypertensive nephropathy, nephrotic syndrome); and non-AIDS-defining cancers. Patients with two or more NCDs were considered as having

multimorbidity. Prevalent comorbidities included diagnoses recorded before or at the start of observation time (first clinic visit for those ≥50 years old at entry and first visit after age 50 for those <50 years old at entry). Incident diagnoses were all non-repeating diagnoses recorded after start of observation period until death, loss to follow-up, or administrative censoring.

## Statistical analysis

We summarized clinical and demographic characteristics of patients ≥50 years, by age at enrollment (<50yo at enrollment and continued in care after 50yo and ≥50yo at enrollment). We compared sociodemographic, clinical and medical history characteristics between these groups to verify and document our assumption that they are different using chi-squared tests for categorical variables and Kruskal-Wallis for continuous variables.

In the first analysis, we estimated four different measures of frequency. First, to describe the temporal changes in the burden of comorbidities, we evaluated the annual prevalence of any single comorbidity and of multimorbidity during the study period (2000–2015) in all patients ≥50 years of age in care that year. Next, to describe the prevalence of NCDs, we used different relevant timepoints in care during individual follow-up (at baseline, during care, and at the end of follow-up) and estimated the prevalence of any NCD, and of multimorbidity. We also estimated the prevalence of specific categories of NCDs at baseline, unadjusted incidence of these during care, and prevalence at end of follow-up.

In the first analysis, we generated an annual cohort of patients ≥50yo with HIV-infection for every calendar year between 2000–2015 with at least two visits each year, one occurring after July 15th. Inclusion criteria guaranteed that patients were actively receiving care in a particular year and had the opportunity of being screened for clinical events during the whole year. In the second analysis, we calculated the proportion of patients with one NCD and with multimorbidity at baseline and at end of the follow-up (accumulated) for each age-at-enrollment group. Any NCD reported at or before start of follow-up was defined as prevalent at baseline, and these and any event diagnosed after baseline were included in the number of comorbidites at end of follow-up. In the third and fourth analyses, we estimated the prevalence of specific NCDs at baseline and at the end of follow-up as described above. We estimated unadjusted incidence of specific NCDs by age-group, excluding prevalent diagnoses, using the number of new registered diagnoses of specific NCDs over the total time of follow-up, censoring observation time at the registered date of the NCD of interest.

## Results

### Cohort characteristics

We included 3,415 patients who were followed for a median of 3.7 years at ages ≥50 years between 2000 and 2015. Of these, 1,928 (56%) were <50 years at enrollment and 1,487 (43%) ≥50 years at enrollment. Median ages at start of follow-up time for this study were 50.5yo and 55.2yo for each group. Patients <50yo at enrollment were followed a median time of 3.1 years after reaching 50 years (cumulative time of 7,821 years in follow-up). In contrast, those ≥50yo at enrollment were followed for 4.6 years (8,064 cumulative years in follow-up). While most patients <50yo at enrollment were receiving ART at start of follow-up time (93%), only 32% of those ≥50yo at enrollment were receiving ART at start of follow-up. Median CD4 counts at start of follow-up were 463 cells/mm$^3$ among people <50yo at enrollment and 246 cells/mL among those ≥50yo at enrollment. While the study period encompasses a 15-year period (2000–2015), 50% of all person time of observation in this study occurred between 2010 and 2015 (Table 1). Most patients were followed-up until 2015 and were administratively censored (79% in patients enrolled< = 50 and 73% in those enrolled ≥50yo). The remaining

**Table 1. Demographic and clinical characteristics at start of follow-up [†] of 3,415 patients 50 years and older receiving HIV care in six centers in Latin America affiliated with CCASAnet, by age at enrollment in HIV care [‡] (2000–2015).**

| Characteristics [§] | <50yo at enrollment in HIV care (n = 1,928) | ≥50yo at enrollment in HIV care (n = 1,487) | Combined (n = 3,415) | p-value |
|---|---|---|---|---|
| **Male** | 1,382 (72%) | 1,094 (73%) | 2,476 (72%) | 0.23 |
| **Age,** (years) | 50.5 (50.2–50.7) | 55.2 (53–60) | 50.8 (50.4–54.6) | <0.001 |
| **CD4 count,** (cells/mm$^3$) [¶] | 463 (302–670) | 246 (98–447) | 373 (196–582) | <0.001 |
| **Probable route of transmission,** n (%) | | | | |
| Heterosexual | 832 (43%) | 693 (47%) | 1,525 (45%) | 0.80 |
| Homosexual | 520 (27%) | 315 (21%) | 835 (24%) | |
| Other | 264 (14%) | 142 (9%) | 406 (9%) | |
| Unknown | 312 (16%) | 337 (23%) | 649 (19%) | |
| **Year of enrollment in care** | 2003 (2001–2007) | 2008 (2005–2012) | 2005 (2002–2009) | <0.001 |
| **Year of enrollment in the cohort** (started follow-up) | 2011 (2008–2013) | 2008 (2005–2012) | 2010 (2006–2013) | <0.001 |
| **Time in follow-up since enrollment in care** (years) | 10.9 (7.7–13.3) | 4.6 (2–8.4) | 8.4 (4.2–12) | <0.001 |
| **Time in follow-up in the ≥50yo cohort** (years) | 3.1 (1.4–6.14) | 4.6 (2–8.4) | 3.7 (1.6–6.9) | <0.001 |
| **Receiving cART at start of follow-up,** n (%) | 1,800 (93%) | 478 (32%) | 2,278 (67%) | <0.01 |
| **Years on cART before start of follow-up** | 6.0(2.8–10.3) | 0 (0–0.4) | 2.9 (0–8) | <0.001 |
| **At least one comorbidity,** n (%) | 818 (42%) | 260 (17%) | 1078 (31%) | <0.001 |
| **Center** [††] n (%) | | | | |
| FH-Argentina | 632 (33%) | 578 (39%) | 1210 (35%) | <0.001 |
| FC-Brazil | 694 (36%) | 461 (31%) | 1155 (34%) | |
| FA-Chile | 355 (18%) | 226 (15%) | 581 (17%) | |
| IHSS/HE-Honduras | 124 (6%) | 97 (6%) | 221 (6%) | |
| INCMNSZ-México | 123 (6%) | 125 (8%) | 248 (7%) | |

† Follow-up started the date of the first visit after their 50[th] birthday for patients <50yo at enrollment and the registered first visit date in the clinic for patients ≥50yo at enrollment.

‡ Patients were dichotomized into two groups according to age at enrollment: people enrolled in care before 50 years old and ageing in care after 50yo (<50y at Enrollment) and patients enrolled in care at 50 years or older (≥50y at Enrollment).

§ Continuous variables are reported as medians (interquartile range).

¶ Missing information on CD4 count for 322 among <50at enrollment and 213 patients ≥50 at enrollment.

†† FH-Argentina (Fundación Huesped), FC-Brazil (Instituto Nacional de Infectología Evandro Chagas, Fiocruz), FA-Chile (Fundación Arriarán), IHSS/HE-Honduras (Instituto Hondureño de Seguridad Social and Hospital Escuela Universitario in Honduras), INCMNSZ-México (Instituto Nacional de Ciencias Médicas y Nutrición Salvador Zubirán)

participants were lost to follow-up (17%) or died (8%). Compared to those ≥50yo at enrollment, a higher proportion of patients <50yo at enrollment were administratively censored (79% vs 73%, respectively) and a lower proportion died (5% vs 11%, respectively).

## Frequency of non-communicable diseases

A total of 3,343 NCDs were reported, of which 1,550 (46%) were diagnosed during follow-up. The proportion of patients ≥50 years of age with at least one NCD increased from 32% to 68% between 2000 and 2015 and those with multimorbidity from 30% to 40% (Fig 1).

At least one NCD at or before starting follow-up was reported in 1,078 (31%) patients.

At the start of study follow-up time, 814 (42%) patients <50yo at enrollment versus 260 (17%) patients ≥50yo at enrollment had one or more NCD. By the end of follow-up the proportion of patients with any NCDs was approximately 50% regardless of whether patients

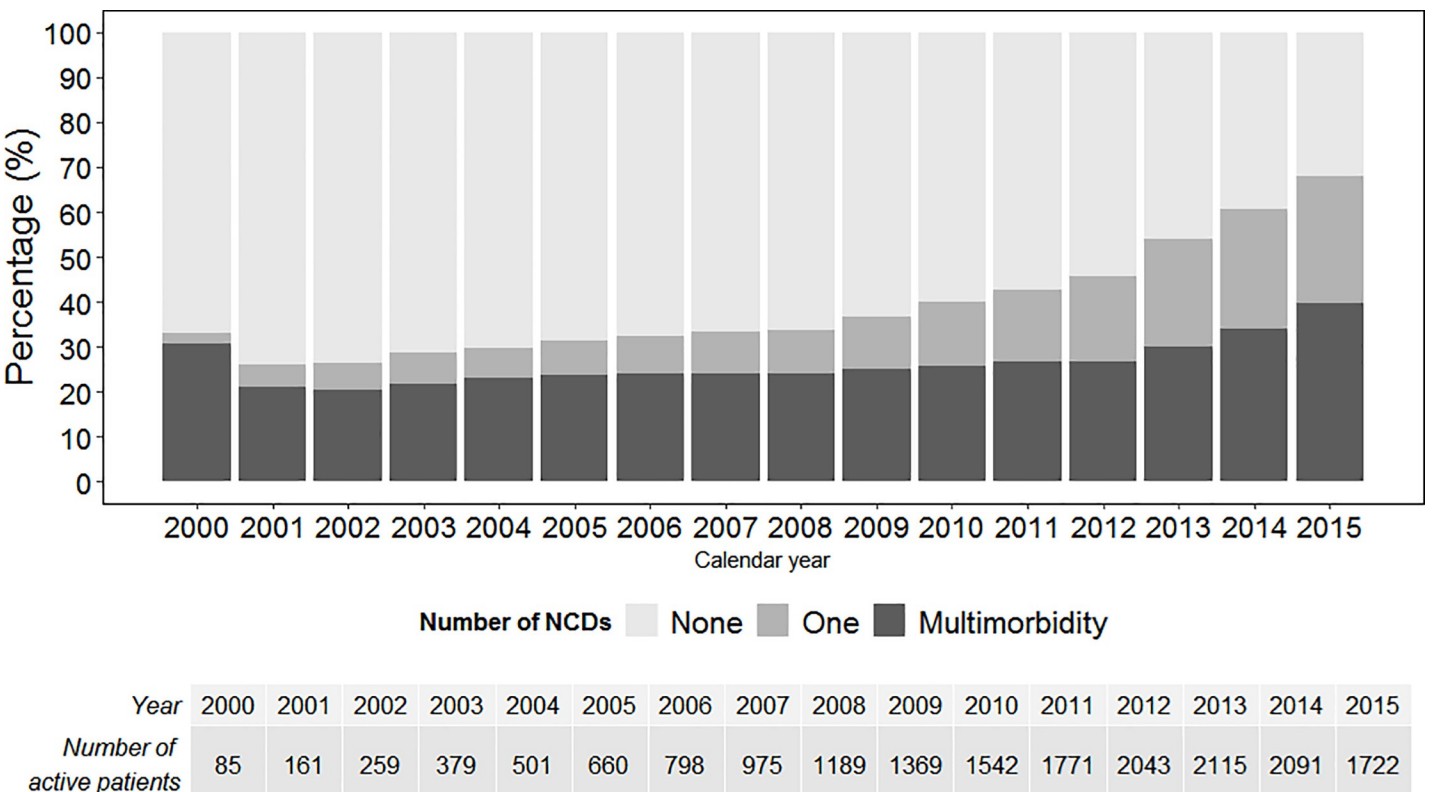

**Fig 1. Annual prevalence of non-communicable diseases (NCDs) and multimorbidity [†] in people 50 years of age or older receiving HIV care in six centers in Latin America affiliated with CCASAnet [§] (2000–2015).** † Multimorbidity defined as having two or more NCDs.

entered clinical care before (53%) or after age 50 years (50%) (Fig 2). The overall incidence of NCDs was 9.75 per 100 person-years of follow-up: 6.31 per 100 person-years (494 NCDs in 7,821 years of follow-up) among those enrolled before 50yo and 13.09 per 100 person-years (1,056 NCDs in 8,064 years of follow-up) among patients enrolled after 50yo.

## Distribution of specific-types of non-communicable diseases

Prevalence and incidence of specific types of NCDs at start and end of follow-up are shown in Table 2. Among patients <50yo at enrollment the most common comorbidities at the start and end of follow-up were dyslipidemia (27% and 36%), hypertension (13% and 17%), psychiatric disorders (11% and 15%), and diabetes (7% and 11%). Similarly, among patients ≥50yo at enrollment, dyslipidemia (3% and 28%), hypertension (8% and 18%), psychiatric disorders (1% and 10%), and diabetes (4% and 12%) were the most frequent disorders at baseline and end of follow-up, respectively. At the end of follow-up, the accumulated proportion of patients with cardiovascular diseases (7% vs. 8%), liver (9% vs. 9%), and renal disease (2% vs. 4%), and non-AIDS-defining malignancy (3% vs. 5%) were similar between patients <50 and ≥50 yo at enrollment. Thus, the estimated incidence of most individual comorbidities was apparently higher among those enrolled after 50 years of age.

## Discussion

This observational, multi-national cohort study shows that a very high proportion of people 50 years of age and older receiving care for HIV in Latin America have non-communicable

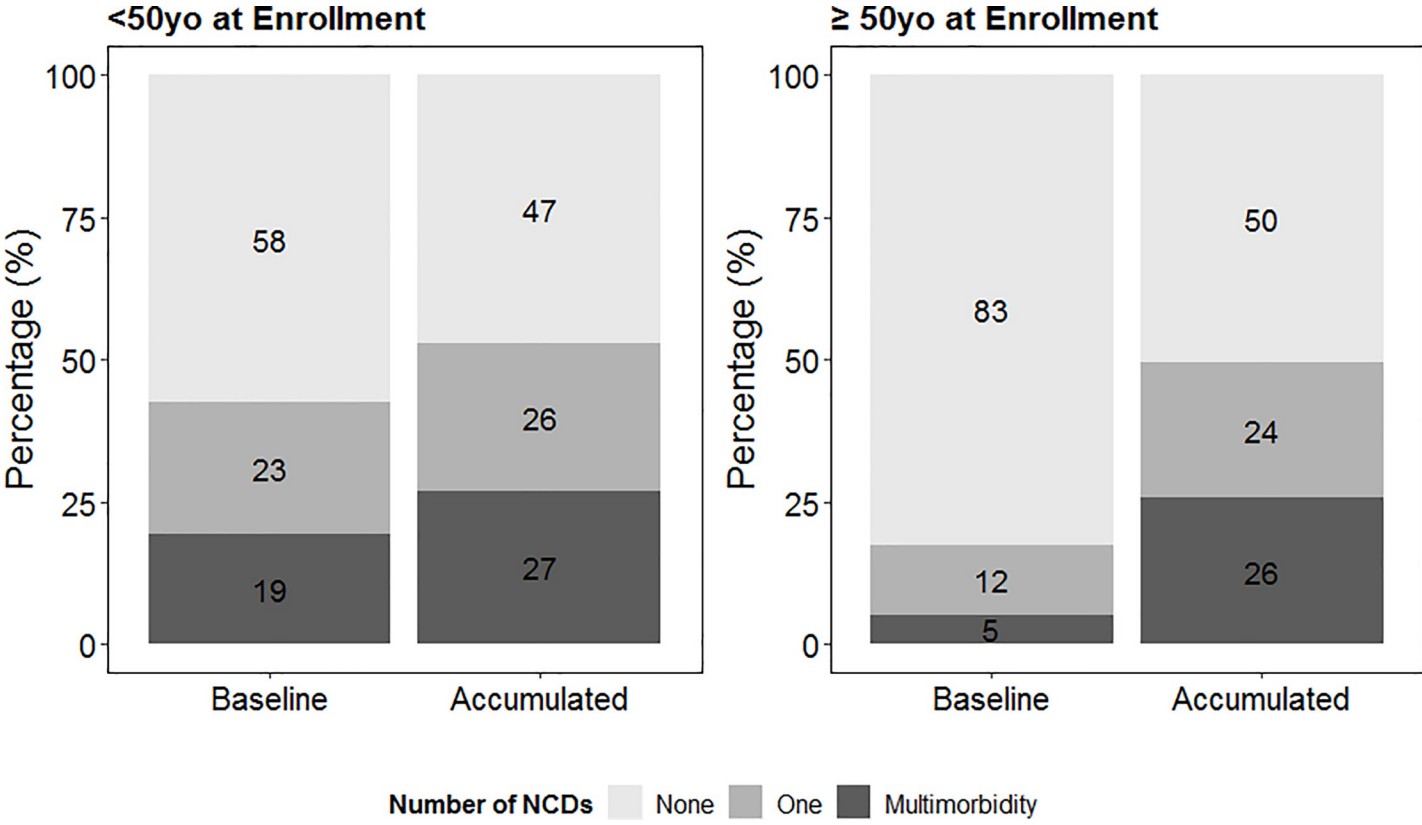

**Fig 2. Prevalence of non-communicable diseases (NCD)[†], and multimorbidity (more than one NCD) at start of follow-up[‡] and at end of follow-up[§] in patients 50 years of age or older receiving HIV care in six centers in Latin America affiliated with CCASAnet [††].** † Including any of the following: diabetes, psychiatric disorders, liver diseases, cardio- and cerebrovascular diseases, renal diseases, and non-AIDS-defining malignancy. ‡ Patients <50 at enrollment started follow-up the date of the first visit after their 50[th] birthday and those ≥50 at enrollment in the date of first registered visit to the clinic. § Prevalence of NCDs and multimorbidity at end of follow-up includes NCDs diagnosed before or at start of follow-up and NCDs diagnosed during follow-up. †† Caribbean, Central and South American network for HIV epidemiology (CCASAnet).

diseases and multimorbidity, and it highlights the increasing need of healthcare models for older people with HIV that integrate prevention and care of NCDs [25,36,37]. We observed that close to 70% of patients ≥50 years in 2015 experienced at least one comorbidity, and half of them two or more. We also observed that at the last individual follow-up visit, around half of patients ≥50yo receiving HIV care in our centers developed at least one NCD and half of these had at least two NCDs, regardless of age at enrollment in care. Dyslipidemia, hypertension, psychiatric disorders, and diabetes were amongst the most frequent NCDs, highlighting also the increasing frequency of risk factors for cardiovascular disorders. Epidemiologic data of NCDs among older PLWHIV in Latin America are notably scarce [21–23]. Thus, this study contributes to fill an important knowledge gap and provides information for better comprehension of healthcare needs of older people with HIV in our region [25, 36–38].

### Frequency of NCDs

We observed a lower prevalence of NCDs at the start of follow-up in patients ≥50yo at enrollment, but the prevalence of any comorbidity and multi-morbidity was similar at the end of follow-up. While people enrolled in HIV care at 50yo or older had an apparently lower prevalence of NCDs at baseline and higher incidence during follow-up, increased opportunities for diagnosis after enrollment in regular medical care for HIV-infection are likely to

**Table 2. Prevalence and incidence † of non-communicable diseases (NCDs) in patients 50 years and older receiving HIV care in six centers in Latin America ‡ affiliated with CCASAnet § by age at enrollment in HIV care (<50yo and ≥50yo at enrollment).**

| Specific non-communicable disorders | <50yo at enrollment (1,928 patients) | | | ≥50yo at enrollment (1,487 patients) | | |
|---|---|---|---|---|---|---|
| | Start of follow-up ¶ (1,437 NCDs) | End of follow-up †† (1,931 NCDs) | Incidence§§ (NCD/100-person-years) | Start of follow-up‡‡ (356 NCDs) | End of follow-up†† (1,412 NCDs) | Incidence§§ (NCD/100-person-years) |
| **Dyslipidemia** | 525 (27%) | 697 (36%) | 2.20 | 52 (3%) | 419 (28%) | 4.69 |
| **Hypertension** | 247 (13%) | 320 (17%) | 0.90 | 115 (8%) | 271 (18%) | 1.93 |
| **Psychiatric** | 222 (11%) | 293 (15%) | 0.91 | 22 (1%) | 148 (10%) | 1.61 |
| **Diabetes** | 136 (7%) | 206 (11%) | 0.87 | 61 (4%) | 183 (12%) | 1.51 |
| **Liver Diseases** | 147 (8%) | 167 (9%) | 0.25 | 50 (3%) | 132 (9%) | 1.05 |
| **Cardiovascular** | 99 (5%) | 141 (7%) | 0.52 | 36 (2%) | 124 (8%) | 1.09 |
| **Malignancies** | 39 (2%) | 64 (3%) | 0.32 | 10 (<1%) | 77 (5%) | 0.86 |
| **Renal Disease** | 22 (1%) | 43 (2%) | 0.26 | 10 (<1%) | 58 (4%) | 0.59 |

† Prevalence when starting follow-up, at end of follow-up, and incidence of specific NCDs.

§ Caribbean, Central and South America network for HIV epidemiology (CCASAnet).

¶ Date of the first visit after their 50th birthday.

†† Date of last registered visit to the center, registered date of diagnosis of the NCD or death.

§§ Number of specific NCDs events newly diagnosed during follow-up over the total number of years of follow-up.

‡‡ Date of first registered visit to the clinic.

account for most of this difference. Antiretroviral associated dyslipidemia might also partially explain these differences. Allmost all patients in our centers were initiated on efavirenz- or boosted lopinavir-based regimens with ziduvudine and lamivudine during the study period [28–30], and dyslipidemia, a common complication of former first-line ART regimens [39], was the most common incident NCD among patients ≥50yo at enrollment. Importantly, more than two-thirds of patients older than 50yo receiving care for HIV in these centers in Latin America, regardless of their age at enrollment, had concomitant NCDs that increase their cardiovascular risk and require integrated management in our clinics.

## Results of the study in context

The prevalence of diabetes, non-AIDS-defining cancer, psychiatric disorders, hypertension, and cardiovascular, renal, and liver diseases in our cohort was lower than in cohorts of older PLWHIV in high-income countries [18–20,40]. Differences in the frequency of environmental, genetic and individual risk factors for NCDs might explain these disparities [41,42].Previous studies in Italy found that people aging while receiving HIV care more frequently had several specific NCDs, multimorbidity, and polypharmacy than people of similar age diagnosed and enrolled in care later in life [18,26]. In contrast, we saw that patients enrolled in care after 50 years of age had a lower prevalence of NCDs than those who reached 50 years of age while in care, whereas those enrolled after 50 years had a higher incidence of NCDs. This discrepancy may potentially be explained by the shorter time in observation for both groups in our study and considerable larger differences of time living with HIV between groups in Italy (≥10 years in comparison with around 5 years in this study), rather than age itself [18,22,26]. In previous, single center studies with a limited number of participants in Brazil (n = 184) and Mexico (n = 208), the overall prevalence of diabetes, dyslipidemia, hypertension, cardiovascular disorders, non-AIDS related neoplasia, chronic non-AIDS related liver and kidney diseases, and of multimorbidity was significantly higher than that at the end of follow-up in our study [22,23]. In contrast, the prevalence of multimorbidity in 2014 was lower than 15% in a large,

multisite, cohort study in Brazil, but the cohort included all adults 18yo and older receiving care, so the mean age of the study population was significantly younger [21].

We observed a lower prevalence of NCDs in our cohort relative to prevalence estimates corresponding to the general population in Latin America. [11–17]. It is unclear whether this is due to incomplete ascertainment of NCDs or selection biases that make a cohort of patients longitudinally receiving treatment and care at a health center not comparable with a representative sample of the general population.

Overall, our study provides robust and generalizable evidence on the frequency of NCDs in older people with HIV by including a large, multinational cohort of older people receiving care for HIV in a diverse group of clinical centers comprising different HIV-care models and populations throughout our region [27–31]. Our results have important implications for HIV care in Latin America as NCDs in PLWHIV, particularly in older populations, significantly reduce survival [10], increase health care costs [4,5,43], and are associated with other geriatric syndromes that increase complexity of care [18,23,44–46], and may reduce HIV therapeutic options [18,46].

## Limitations

The scope of our study is limited to the initial identification and description of the problem and has a number of important limitations to consider. First, we are unable to provide information on relevant risk factors for NCDs as CCASAnet does not routinely collect data on diet, physical activity, body composition, or tobacco use. Similarly, we focused on NCDs and multi-morbidity as measures of morbidity; nonetheless, other unmeasured variables (e.g., disability, polypharmacy, neurocognitive decline, and other geriatric syndromes) might be more relevant to health-related quality of life and healthcare needs in aging people [23, 43–47]. In future studies, we plan to address these limitations with prospective data collection of relevant risk factors and clinical assessments for geriatric syndromes, including data on people of similar age without HIV-infection. Further epidemiological, clinical and policy research is needed to specifically address these implications in the region [4,5,25,36–38,47].

We acknowledge several other methodological limitations in our work, such as the uncertainty about the representativeness of CCASAnet participating centers of routine HIV-care in Latin America. As with any other observational study, our estimates might be biased due to non-participation and attrition, and other of several biases arising from factors which might include differences in participation in care and quality of care at each center. We classified our study population in two groups based on their age at enrollment in care. We acknowledge this is an arbitrary distinction that does not necessarily accurately depict a complex aging biological process intertwined with the effects of long-term toxicity of ART. Nonetheless, ours is a descriptive study aiming to quantify this emerging problem and we found this a useful solution to describe separately subgroups in our population of interest that present different morbidity patterns. Despite systematic efforts to improve data quality and management in CCASAnet [33,34], this data is subject to potential information biases as NCDs diagnoses were included solely if recorded in the medical record by the provider. Clinical file management practices may have differed by site and over time, and screening for NCDs is not standardized across centers [48]. Health care seeking behavior also affects recognition and capture of diagnoses for these conditions. Thus, underdiagnosis and under-reporting of NCDs might have led us to underestimate the real frequency of these disorders [49–51].

Despite these limitations, our study provides very much needed information that may guide the development of specific guidelines of integrated health care for the older, adult HIV population, as currently only a few countries offer recommendations for preventive, integrated primary care for older people living with or without HIV in our region [52–55].

## Conclusions and potential impact

Our findings have important implications for planning healthcare services for older people living with HIV in Latin America. The high prevalence of comorbidities and multimorbidity among older people living with HIV in our region should be taken into consideration to plan for the provision of integrated care [17,30]. Increasing age and frequencies of NCDs in this population may increase costs for primary prevention, screening, care and treatment of NCDs [4, 5, 40–44], and have an impact on the quality of care and choices of available cART [43]. Furthermore, NCDs also increase mortality in patients with complete virologic suppression and adequate immune recovery receiving cART, underscoring, the importance of identifying the frequency of non-AIDS associated risk factors for mortality of older people living with HIV [10, 41].

## Acknowledgments

The Caribbean, Central and South American network for HIV epidemiology (CCASAnet) includes research teams in the following sites: Fundación Huésped, Argentina: Pedro Cahn, Carina Cesar, Valeria Fink, Omar Sued, Emanuel Dell'Isola, Cleyton Yamamoto, Florencia Cahn; Instituto Nacional de Infectologia-Fiocruz, Brazil: Beatriz Grinsztejn, Valdilea Veloso, Paula Luz, Raquel de Boni, Sandra Cardoso Wagner, Ruth Friedman, Rodrigo C. Moreira; Universidade Federal de Minas Gerais, Brazil: Jorge Pinto, Flavia Ferreira, Marcelle Maia; Universidade Federal de São Paulo, Brazil: Regina Célia de Menezes Succi, Daisy Maria Machado, Aida de Fátima Barbosa Gouvêa; Fundación Arriarán, Chile: Marcelo Wolff, Claudia Cortes, Maria Fernanda Rodriguez, Gladys Allendes; Les Centres GHESKIO, Haiti: Jean William Pape, Vanessa Rouzier, Adias Marcelin, Christian Perodin; Hospital Escuela Universitario, Honduras: Marco Tulio Luque. Instituto Hondureño de Seguridad Social, Honduras: Denis Padgett. Instituto Nacional de Ciencias Médicas y Nutrición Salvador Zubirán, Mexico: Juan Sierra Madero, Brenda Crabtree Ramirez, Pablo F Belaunzaran, Yanink Caro Vega. Instituto de Medicina Tropical Alexander von Humboldt, Peru: Eduardo Gotuzzo, Fernando Mejia, Gabriela Carriquiry. Vanderbilt University Medical Center, USA: Catherine C McGowan, Bryan E Shepherd, Timothy Sterling, Karu Jayathilake, Anna K Person, Peter F Rebeiro, Mark J Giganti, Jessica Castilho, Stephany N Duda, Fernanda Maruri, Hilary Vansell.

## Author Contributions

**Conceptualization:** Pablo F. Belaunzaran-Zamudio, Yanink Caro-Vega, Jessica L. Castilho, Bryan E. Shepherd, Carina Cesar, Catherine C. McGowan, Juan G. Sierra-Madero.

**Data curation:** Yanink Caro-Vega.

**Formal analysis:** Pablo F. Belaunzaran-Zamudio, Yanink Caro-Vega, Mark J. Giganti, Bryan E. Shepherd.

**Funding acquisition:** Catherine C. McGowan, Juan G. Sierra-Madero.

**Investigation:** Pablo F. Belaunzaran-Zamudio, Yanink Caro-Vega, Mark J. Giganti, Jessica L. Castilho, Brenda E. Crabtree-Ramirez, Bryan E. Shepherd, Fernando Mejía, Carina Cesar, Rodrigo C. Moreira, Marcelo Wolff, Jean W. Pape, Denis Padgett, Catherine C. McGowan, Juan G. Sierra-Madero.

**Methodology:** Pablo F. Belaunzaran-Zamudio, Yanink Caro-Vega, Mark J. Giganti, Jessica L. Castilho, Bryan E. Shepherd, Fernando Mejía, Rodrigo C. Moreira, Marcelo Wolff, Jean W. Pape, Denis Padgett, Catherine C. McGowan, Juan G. Sierra-Madero.

**Project administration:** Catherine C. McGowan.

**Resources:** Catherine C. McGowan, Juan G. Sierra-Madero.

**Supervision:** Pablo F. Belaunzaran-Zamudio, Yanink Caro-Vega, Jessica L. Castilho, Brenda E. Crabtree-Ramirez, Bryan E. Shepherd, Carina Cesar, Rodrigo C. Moreira, Marcelo Wolff, Jean W. Pape, Denis Padgett, Catherine C. McGowan, Juan G. Sierra-Madero.

**Validation:** Pablo F. Belaunzaran-Zamudio, Yanink Caro-Vega, Jessica L. Castilho, Brenda E. Crabtree-Ramirez, Bryan E. Shepherd, Fernando Mejía, Carina Cesar, Rodrigo C. Moreira, Marcelo Wolff, Jean W. Pape, Denis Padgett, Catherine C. McGowan, Juan G. Sierra-Madero.

**Writing – original draft:** Pablo F. Belaunzaran-Zamudio, Yanink Caro-Vega, Mark J. Giganti, Jessica L. Castilho, Brenda E. Crabtree-Ramirez, Bryan E. Shepherd, Fernando Mejía, Carina Cesar, Rodrigo C. Moreira, Marcelo Wolff, Jean W. Pape, Denis Padgett, Catherine C. McGowan, Juan G. Sierra-Madero.

**Writing – review & editing:** Pablo F. Belaunzaran-Zamudio, Yanink Caro-Vega, Mark J. Giganti, Jessica L. Castilho, Brenda E. Crabtree-Ramirez, Bryan E. Shepherd, Fernando Mejía, Carina Cesar, Rodrigo C. Moreira, Marcelo Wolff, Jean W. Pape, Denis Padgett, Catherine C. McGowan, Juan G. Sierra-Madero.

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
