## [Decision Letter · Decision Letter 0]

6 Nov 2019

PONE-D-19-27737

Frequency of non-communicable diseases in people 50 years of age and older receiving HIV care in Latin America.

PLOS ONE

Dear PhD Caro-Vega,

Thank you for submitting your manuscript to PLOS ONE. After careful consideration, we feel that it has merit but does not fully meet PLOS ONE’s publication criteria as it currently stands. Therefore, we invite you to submit a revised version of the manuscript that addresses the points raised during the review process.

Two reviewers have assessed the manuscript. Their comments are appended below. The reviewers have raised significant concerns about the manuscript, and in particular, they feel that some methodological issues exist (especially data analysis) that affect the technical soundness of your study. A data reanalysis was proposed, and authors should forward adequately. Furthermore, The discussion section requires several improvements, including additional arguments of the implications of the findings, comparisons with other studies and limitations of the manuscript.  

We would appreciate receiving your revised manuscript by Dec 21 2019 11:59PM. To enhance the reproducibility of your results, we recommend that if applicable you deposit your laboratory protocols in protocols.io, where a protocol can be assigned its own identifier (DOI) such that it can be cited independently in the future. For instructions see: http://journals.plos.org/plosone/s/submission-guidelines#loc-laboratory-protocols

We look forward to receiving your revised manuscript.

Kind regards,

Bruno Pereira Nunes, Ph.D.

Academic Editor

PLOS ONE

Journal Requirements:

1.  Thank you for including your ethics statement: "Institutional review board approval was obtained locally for each participating site and for the DCC at Vanderbilt. In each of the sites contributing data to this study, ethical regulations and policies permit retrospective analysis of de-identified clinical data without informed consent when research is approved by an Institutional Review Board or appropriately constituted ethics committee. "

2. Thank you for including your competing interests statement; "Pablo F Belaunzaran-Zamudio, Yanink Caro-Vega, Mark J Giganti, Brenda Crabtree-Ramírez, Bryan E Shepherd, Carina Cesar, Rodrigo C Moreira, Fernando Mejia, Marcelo Wolff, Jean W. Pape, Denis Padgett and Catherine C McGowan have no conflicts to declare. Juan Sierra-Madero reports personal fees and non-financial support from Gilead, non-financial support from MSD, grants from BMS, grants from Pfizer, and personal fees from Jansen, all outside the submitted work. "

Additional Editor Comments (if provided):

- Avoid abbreviations, if possible

Reviewers' comments:

Reviewer's Responses to Questions

**Comments to the Author**

1. Is the manuscript technically sound, and do the data support the conclusions?

Reviewer #1: Partly

Reviewer #2: Yes

2. Has the statistical analysis been performed appropriately and rigorously? 

Reviewer #1: Yes

Reviewer #2: Yes

3. Have the authors made all data underlying the findings in their manuscript fully available?

Reviewer #1: No

Reviewer #2: No

4. Is the manuscript presented in an intelligible fashion and written in standard English?

Reviewer #1: Yes

Reviewer #2: Yes

5. Review Comments to the Author

Reviewer #1: Non-communicable diseases and the accumulation of them within PLWH is of significant and growing concern given an expected increase in the number of PLWH aged >50 years of age. As authors note, there are a limited number of studies quantifying the epidemiology of NCDs and multimorbidity in a Latin American setting, particularly in large numbers. This important study provides evidence of a growing prevalence of NCDs, but would be strengthened by refining study objectives, clarifying methods and implications of findings.

Major Comments

1. Authors conclude that regardless of age at enrollment, study findings suggest that a high prevalence of NCDs and multimorbidity were observed. How are these results reconciled with their statement that planning and provision of complex, primary care for PLWH should target mainly those who are 50+?

2. To be included in this observational cohort analysis, patients being observed for clinical care need to contribute at least 2 clinical visits, at least one of which occurred during the study period of 2000-2015 (line 103). How are new or “incident” events determined if only one visit was available? Did this analysis take place within a subset of the full study population of n=3415?

3. In the discussion, authors note that a limitation of their study is that events may have been underascertained. Given the inclusion of multiple study sites and the potential for heterogenous data collection (ie the point in time during which data was collected may vary by site), did authors have a systematic way of ensuring that all constituent NCDs had data available during overlapping periods of time? For example, to assess whether a person had X of 8 proposed categories of NCDs, presumably all 8 categories of NCDs were being collected at the same time.

4. As visit structure of participants is driven by clinical care seeking behavior, “incident” NCDs may not represent true biological onset and should be addressed in the limitations. Relatedly, as in line 156, prevalent cases were removed at or before the start of follow-up. It would be helpful to clarify whether there was window of time during which events were considered “prevalent” (for example, 6 months before and after enrollment).

5. In line 231, authors discuss results presented in Table 2. More specifically, that incidence rates of NCDs were higher among PLWH who enrolled in care when they were >50 years old, as compared with individuals who were <50 years old at enrollment. Presumably, to adjust for age “patients <50 at enrollment started follow-up at their first visit after their 50th birthday” (line 146-47), however, are the groups comparable if those <50 years at enrollment may have experienced incident NCDs at an earlier age during the course of their clinical care, simply not captured by the study period?

6. The denominator for prevalence was “all older patients older than 50 with at least 1 visit after July 15th”. Please provide rationale.

7. In line 155, is “start and end of follow-up” all time after 50 years of age up until death/NCD event/2015?. Similarly, in line 161, is “total time of follow-up” = end of follow up – start of follow up?

Minor Comments

1. The abstract’s background reads as though authors intended to quantify frequency of NCDs, individual and in aggregate, among PLWH older than 50. However, it seems their objective to describe the rate at which they occur (incidence) was excluded.

2. In Line 83, “multimorbidity was the norm in both centers”. Please quantify and cite.

3. The outcomes of NCDs appear to be a blend of “registered” and “reported” events. Is this distinction referring to whether outcomes were self-reported vs physician diagnosed?

4. Please provide citations for the statement in line 270 “differences in the frequency of environmental, genetic, and individual risk factors for NCDs might explain disparities”.

5. Statistical tests in Table 1 were performed, but not described in the Methods’ statistical analysis plan.

Reviewer #2: This paper aims to estimate the prevalence of NCDs amongst PLHIV in Latin America using CCASAnet cohort data. This is a clear, well written and timely study which adds important data to a field that is gaining increasing momentum. Particular value is the region these findings are reporting on, with Latin America being a region were such data is lacking compared to other regions such as North America and Europe and even SSA.

My main concern centre around the choice of sub-analysis and its focus on age at enrolment only instead of maximising use of this large dataset to e.g. generate regression analysis looking at risk factors for NCDs. See detailed comments below

Detailed comments:

- It is important to distinguish the impact of aging PLHIV on HIV care and aging in general on the health system (including primary care). HIV remains a low prevalent disease and thus is unlikely to have a marked impact on health systems more generally compared to the aging general population. The abstract and introduction should be updated accordingly.

- Stratifying by age at enrolment seems an odd choice of sub-analysis as a simple descriptive overall analysis would already provide important results. As stated by authors in introduction ART exposure, previous health history, HIV duration and severity could drive NCD burden beyond age and may drive different NCDs. A sub-analysis by age at enrolment is not adequate to control for these factors, as hinted in introduction. I assume the CCASAnet cohort provides data on some of these factors and other (e.g. smoking, BMI, etc). Would the authors not consider dissecting this issue of NCD further by looking at risk factors for NCDs as other studies from HIC have done using regression to tease out effect better? This seems to be a major limitation of the paper and such addition would provide a wealth of additional data to this paper. If this is not feasible this would fully discussed in discussion

- Authors should provide a little more information on the CCASAnet cohort for the reader. Including that it compromises clinical data from 7 countries, how NCDs are defined (e.g. are clinical definitions the same across facilities, if so provide these (e.g. is hypertension 140/90 or 130/80?), if not this should be stated and mentioned in discussion as limitation)

- Authors should explain why their analysis is limited to data from only 5 countries and not all 7 countries in CCASAnet. I assume this is because some facilities did not report on NCDs?

- How was enrolment defined? HIV diagnosis? At ART initiation? Other?

- NCD prevalence is higher in <50 but by end of follow-up it was the same. Authors should dissect this somehow to better help the reader understand this difference at the beginning. Authors should hint at an explanation in result section and elaborate in discussion. The same comment applies when reporting on individual NCDs in the next section. Why dyslipidaemia for example is 27% in <50s and 3% in over 50s).

- As pointed out the lower prevalence in this study compared to other settings may be explained by lower screening? What do guidelines say about frequency of screening in PLHIV, is there data on gap between guidelines and real world (e.g. as done in https://www.ncbi.nlm.nih.gov/pubmed/28578613 for the Netherlands)? Could they also provide data on eg prevalence of hypertension in HIV-negative from both regions to support the hypothesis of genetic/risk factor does result in lower prevalence in Latin America?

- The discussion is somewhat short. Authors should elaborate on a number of points:

o How do the results compare to other data from the region and beyond show eg the Brazil and Mexico data mentioned in intro

o What are the implications for clinical care (prevention, screening, guidelines, polypharmacy eg Smit et al Lancet ID 2015)

o What gaps remain (e.g. more information on proportion of those screened who are treated and controlled) and what data would need to be collected. This is a great opportunity to highlight not just a huge burden but also call for more research – use it �

o How generalisable are the results given that it includes only 5 countries from the region. E.g. how different are some regions in Latin America.

o How much data do Mexico and Brazil contribute to CCASAnet data (not the main contributors when looking at numbers overall but not sure for >50yo). As the previous study into NCD from the region included data from Mexico and Brazil it would be good for authors to clearly highlighted added benefits of this study in discussion in relation to existing evidence (multi-cohort, larger number, more generalisable to whole region etc).

o Would it be useful to have cohorts with controls as in US and Europe to better understand the excess risk of HIV disease and ART on NCDs

General comments

- Authors could use epidemiological terminology e.g prevalence instead of frequency, proportion etc

- When reporting on which NCDs contribute the largest burden, authors could focus on reporting the 3 main NCDs. The number of ‘main’ NCDs reported varies between abstract, results and discussion.

- For figures authors may consider using increasingly darker shades from 0 to multimorbidity instead of a darker colour in the middle for 1NCD? If staying in grey shades and increasing in darkness makes it tricky to read consider using patterns (e.g. no pattern=0 NCDs, light pattern is 1 and dark pattern is 2+)

-

Minor comments

- In abstract could authors be more explicit in abstract that NCD prevalence was estimated amongst PLHIV (not general population attending centres)

- In conclusions of abstract authors may also like to elude to the fact that the burden seems to be increasing over time

- In introduction could authors state that the study was “from Name et al” instead of “from our colleagues” colleagues?

- In introduction when reporting on reasons for excess mortality authors should say “possible reason could be the increased presence” instead of “is”

- Discussion could benefit from use of paragraph to distinguish individual sections.

- In discussion authors state that HTN, dyslipidaemia, psychiatric disorders AND diabetes are the main NCDs but diabetes was not mentioned in result or abstract.

- It would strengthen discussion to mention what drives observed trends when discussion difference in NCD burden at enrolment and end of follow-up (higher incidence)

- I am not sure I understand the sentence “Another potential limitation is that the prevalence …”

- The sentence “As such, we focus… “is misleading given the point on underdiagnosis. Authors may want instead to comment on gaps could be addressed for future analysis (analysis of screening practice to see if it is true underdiagnosis, that underscreening compared to guidelines has also been reported in HIC eg https://www.ncbi.nlm.nih.gov/pubmed/28578613.

6. PLOS authors have the option to publish the peer review history of their article (what does this mean?). If published, this will include your full peer review and any attached files.

Reviewer #1: No

Reviewer #2: No

---

## [Author Response · Author response to Decision Letter 0]

20 Jan 2020

Mexico City, January 20th, 2020

Bruno Pereira Nunes, Ph.D.

Academic Editor

PLOS ONE

Dear Dr Pereira.

We appreciate that a revised version of our manuscript is being considered for publication at PLOS ONE. We also appreciate the thorough review and thoughtful comments by our peers. We are submitting a revised version of the manuscript that intends to address the points raised during the review process. In the next pages of this document we provide a response to each point, indicating the corresponding changes in the manuscript. We are also submitting two versions of the revised manuscript: in one version we highlighted all changes we did to our originally submission and a second version with all changes accepted. 

Responses to Reviewers:

Reviewer #1: 

Non-communicable diseases and the accumulation of them within PLWH is of significant and growing concern given an expected increase in the number of PLWH aged >50 years of age. As authors note, there are a limited number of studies quantifying the epidemiology of NCDs and multimorbidity in a Latin American setting, particularly in large numbers. This important study provides evidence of a growing prevalence of NCDs, but would be strengthened by refining study objectives, clarifying methods and implications of findings.

Major Comments

1. Authors conclude that regardless of age at enrollment, study findings suggest that a high prevalence of NCDs and multimorbidity were observed. How are these results reconciled with their statement that planning and provision of complex, primary care for PLWH should target mainly those who are 50+?

ANSWER: All patients included in the study were 50 years and older at the start of observation time for this study. We limited application of conclusions from the study are to people older than 50yo. 

We made changes at the Introduction and Methods sections that we think will help to better understand the purpose this analysis. Our overall goal was to provide further information on the frequency of comorbidities in people ≥50 years of age receiving care for HIV in our region, and contribute to a better understanding of the magnitude of this problem. This information might be used to inform planning care, and estimate costs and needs of HIV-related healthcare in Latin America, now that the population of people living with HIV is aging.

2. To be included in this observational cohort analysis, patients being observed for clinical care need to contribute at least 2 clinical visits, at least one of which occurred during the study period of 2000-2015 (line 103). How are new or “incident” events determined if only one visit was available? Did this analysis take place within a subset of the full study population of n=3415?

ANSWER: Per inclusion criteria, there were no patients with only one visit available. Patients were included if they had at least 2 clinical visits to ensure all patients contributed observation time. Patients with only one clinic visit were excluded. We performed three separate analysis; inclusion criteria were identical for the whole study population (n=3,415). 

3. In the discussion, authors note that a limitation of their study is that events may have been underascertained. Given the inclusion of multiple study sites and the potential for heterogenous data collection (ie the point in time during which data was collected may vary by site), did authors have a systematic way of ensuring that all constituent NCDs had data available during overlapping periods of time? For example, to assess whether a person had X of 8 proposed categories of NCDs, presumably all 8 categories of NCDs were being collected at the same time.

ANSWER: All 8 categories of NCDs were being collected at the same time in all sites during the whole study period, per cohort design (see McGowan CC, et al. Int J Epidemiol. 2007; reference 19). The cohort has standardized data collection procedures and quality assurance methods. We added a description of these in the subsection “Cohort setting and procedures” and references (pages 5-6, lines 116-143; references 19-26) in the new version of the manuscript. However, screening for NCDs is not standardized across all clinical sites and might have varied by center over time. We have added this acknowledgement to the limitations in the Discussion section (page 19 lines 384-391). We also expanded the discussion of this potential limitation, adding references that support the systematic under-ascertainment of NCDs in adults receiving care for HIV during routine care (references #25, #26 and #39).

4. As visit structure of participants is driven by clinical care seeking behavior, “incident” NCDs may not represent true biological onset and should be addressed in the limitations. Relatedly, as in line 156, prevalent cases were removed at or before the start of follow-up. It would be helpful to clarify whether there was window of time during which events were considered “prevalent” (for example, 6 months before and after enrollment).

ANSWER: We agree with the reviewer’s observation that health care seeking behavior affects recognition and capture of diagnoses for these conditions. We acknowledged this in the Discussion (see page 19, lines 388-391). We don’t know the exact biologic onset of many NCDs (cancers, diabetes, heart disease, etc.) and clinically we use date of diagnosis as disease onset, recognizing this limitation in medicine. Using date of diagnosis as event is consistent with clinical practice. 

Prevalent comorbidities included diagnoses recorded before or at the start of observation time (first clinic visit for those ≥50 years old at entry or first visit after age 50 for those <50 years old at entry). Incident diagnoses were all non-repeating diagnoses recorded after start of observation period until death, loss to follow-up, or administrative censoring. We have clarified these definitions in the Methods section (see page 8, lines 185-189). 

5. In line 231, authors discuss results presented in Table 2. More specifically, that incidence rates of NCDs were higher among PLWH who enrolled in care when they were >50 years old, as compared with individuals who were <50 years old at enrollment. Presumably, to adjust for age “patients <50 at enrollment started follow-up at their first visit after their 50th birthday” (line 146-47), however, are the groups comparable if those <50 years at enrollment may have experienced incident NCDs at an earlier age during the course of their clinical care, simply not captured by the study period?

ANSWER: In our view, these two groups are not comparable. The reason for presenting the groups by age at enrolment is not to attempt to adjust for age, which would not be appropriate through stratification. Our purpose was to describe separately two groups with distinctive clinical histories, so we did no formal comparison between groups.

In this new version, we modified the text and avoid statements suggesting that we are comparing incidence of diseases between these groups. We also modified the language in the Methods (See page 7, lines 163-166), Results (see page 16, lines 328-331) and Discussion (see page 16, lines 315-320, page 17, 336-339) to address this comment. 

We did substantial modifications to the Methods section, to address this and other comments by both reviewers. This including extensive re-writing to provide clearer and more explicit definitions. We did not include in this response all changes to this letter of response, but reviewers will be able to track them easily as all changes are identified with the track-changes function in Word. We hope that the new version reflects more accurately our thinking during the analysis.

Finally, had those <50 years at enrollment been diagnosed with NCDs at an earlier age during the course of their medical care, these were recorded as “prevalent” non-communicable disorders at cohort entry.

6. The denominator for prevalence was “all older patients older than 50 with at least 1 visit after July 15th”. Please provide rationale.

ANSWER: In the analysis to estimate the annual prevalence of comorbidities (summarized in Figure 1), patients were included if had at least a clinical visit after July 15th in the year analysed to ensure they were retained and actively receiving care that year. We modified the description of the analysis to provide a clearer description of this selection (page 9, lines 213-217).

7. In line 155, is “start and end of follow-up” all time after 50 years of age up until death/NCD event/2015?. Similarly, in line 161, is “total time of follow-up” = end of follow up – start of follow up?

ANSWER: This is correct.

Minor Comments

1. The abstract’s background reads as though authors intended to quantify frequency of NCDs, individual and in aggregate, among PLWH older than 50. However, it seems their objective to describe the rate at which they occur (incidence) was excluded.

ANSWER: Thank you for this observation. We have corrected the Abstract's background to include incidence (page 2, lines 42-45).

2. In Line 83, “multimorbidity was the norm in both centers”. Please quantify and cite.

ANSWER: We modified this sentence as recommended. The text in the new version states: “Moreover, the mean number of comorbidities was 1.4 in Mexico and 2 in Brazil, with an overall prevalence of multimorbidity of 63% in Brazil [15,16].”. References have been added (page 4, lines 86-88, References #15 and #16)

3. The outcomes of NCDs appear to be a blend of “registered” and “reported” events. Is this distinction referring to whether outcomes were self-reported vs physician diagnosed?

ANSWER: We used the terms “registered” and “reported” for any event in the dataset without distinction. In the revised version, we consistently use the term “registered” for all events to avoid confusion. Each center collect data during routine HIV-care and might register a new non-communicable disease if this was diagnosed in the center since their last HIV-care visit, if the patient self-reported to have been diagnosed with a new NCDs since their last HIV-care visit elsewhere or if in that visit, their HIV-care physician diagnosed a new NCD. This data is collected from the clinical files and medical notes. 

4. Please provide citations for the statement in line 270 “differences in the frequency of environmental, genetic, and individual risk factors for NCDs might explain disparities”.

ANSWER: We added the references #32 and #33. Please see page 17, lines 344-346. 

5. Statistical tests in Table 1 were performed, but not described in the Methods’ statistical analysis plan.

ANSWER: Thanks for pointing this out. We added the description in the methods section (page 9, lines 201-204), as follows: 

“We compared sociodemographic, clinical and medical history characteristics between these groups to verify and document our assumption that they are different using chi-squared tests for categorical variables and Kruskal-Wallis for continuous variables.”

Reviewer # 2

This paper aims to estimate the prevalence of NCDs amongst PLHIV in Latin America using CCASAnet cohort data. This is a clear, well written and timely study which adds important data to a field that is gaining increasing momentum. Particular value is the region these findings are reporting on, with Latin America being a region were such data is lacking compared to other regions such as North America and Europe and even SSA. 

My main concern centre around the choice of sub-analysis and its focus on age at enrolment only instead of maximising use of this large dataset to e.g. generate regression analysis looking at risk factors for NCDs. See detailed comments below

Detailed comments:

- It is important to distinguish the impact of aging PLHIV on HIV care and aging in general on the health system (including primary care). HIV remains a low prevalent disease and thus is unlikely to have a marked impact on health systems more generally compared to the aging general population. The abstract and introduction should be updated accordingly. 

ANSWER: We completely agree with this comment. We modified the Abstract and the main text Background sections to accurately reflect this. The new version states: “A growing population of older adults with HIV will increase demands on HIV-related healthcare.” (page 2, lines 39-40) and “As the population of PLWHIV ages, co-morbidities resulting from natural aging, effects of chronic inflammation, and long-term use of cART are expected to also grow, increasing demands on HIV-related health care [5,6].” (page 3, lines 68-71).

- Stratifying by age at enrolment seems an odd choice of sub-analysis as a simple descriptive overall analysis would already provide important results. As stated by authors in introduction ART exposure, previous health history, HIV duration and severity could drive NCD burden beyond age and may drive different NCDs. A sub-analysis by age at enrolment is not adequate to control for these factors, as hinted in introduction. 

ANSWER: We agree that stratifying by age at enrollment does not adequately control for differences driving NCD in these two groups. In our view, these two groups are not comparable. We elaborate further in response to commentary #5 of reviewer 1. Briefly, we are presenting the groups by age at enrolment to describe separately two groups of our interest that are likely not part of the same population, and there is no simple method to adjust for their differences. We acknowledge the language we used in the abstract and main text does not conveys our reasoning clearly, so we modified different sections throughout the manuscript for clarity purposes (see page 7 lines 163-166, page 16, lines 328-331; and page 16, lines 315-320 and page 17, 336-339 in the Methods, Results and Discussion sections)

- I assume the CCASAnet cohort provides data on some of these factors and other (e.g. smoking, BMI, etc). Would the authors not consider dissecting this issue of NCD further by looking at risk factors for NCDs as other studies from HIC have done using regression to tease out effect better? This seems to be a major limitation of the paper and such addition would provide a wealth of additional data to this paper. If this is not feasible this would fully discussed in discussion.

ANSWER: CCASAnet cohort does not routinely collect data on some of these other primary risk factors for NCDs (e.g., smoking). The goal of this paper was to focus on the growing population of PLWH ≥50 years and the burden of their other medical comorbidities. Evaluation of NCD risk factors is the next step to identify who is at greatest risk and identify modifiable risk factors for the development of NCDs (regardless of age) in the region. An important but different question that we agree needs further research. We have added some sentences related to this in the discussion, please see page 18, lines 376-377 and page 19, lines 381-384.

- Authors should provide a little more information on the CCASAnet cohort for the reader. Including that it compromises clinical data from 7 countries, how NCDs are defined (e.g. are clinical definitions the same across facilities, if so provide these (e.g. is hypertension 140/90 or 130/80?), if not this should be stated and mentioned in discussion as limitation)

ANSWER: We modified substantially the Methods section to address this and other comments by reviewers. In the new version, we included a new sub-heading titled: “Cohort settings and procedures” where we describe how data on NCD is defined and collected (pages 5-6, see lines 116-143). In the new version, we also address the limitations of our definitions and collection methods (see pages 18-19, lines 370-395). 

- Authors should explain why their analysis is limited to data from only 5 countries and not all 7 countries in CCASAnet. I assume this is because some facilities did not report on NCDs? 

ANSWER: This is correct. We included data from sites where information on non-communicable disorders is available (6 centers in 5 countries). We added in the Methods section (see pages 6-7, lines 147-152) a statement to clarify this point: “We included data of patients receiving care at the six centers where data on non-communicable diseases are collected (Fundación Huesped in Argentina, Instituto Nacional de Infectologia Evandro Chagas -Fiocruz in Brazil, Fundación Arriarán in Chile, Instituto Hondureño de Seguridad Social and Hospital Escuela Universitario in Honduras, and Instituto Nacional de Ciencias Médicas y Nutrición Salvador Zubirán in Mexico).” 

- How was enrolment defined? HIV diagnosis? At ART initiation? Other?

ANSWER: CCASAnet have standardized definitions for these variables. Details and definitions are described in reference #19 and the set of variables are publicly available in documents accessible through: https://www.ccasanet.org

For the reviewer, definitions according to these documents are:

Enrollment: Date of enrolment in HIV care in each site.

HIV diagnosis: Date of first HIV positive test.

ART initiation: date of first ART regimen. 

NCD prevalence is higher in <50 but by end of follow-up it was the same. Authors should dissect this somehow to better help the reader understand this difference at the beginning. Authors should hint at an explanation in result section and elaborate in discussion. The same comment applies when reporting on individual NCDs in the next section. Why dyslipidaemia for example is 27% in <50s and 3% in over 50s). 

ANSWER: We have added some sentences to explain our interest to describe separately two groups with distinctive clinical histories and the potential reasons of those differences. Please see our response for the comment 5 of reviewer 1 and Methods (pages 7, lines 163-166) and Results section (page 16, lines 328-331).

In brief, we attribute differences in healthcare seeking behaviors and access as the main contributor to differences in baseline prevalence of any NCDs and multimorbidity. After enrollment in care, we suspect that greater opportunities of screening and diagnosis of NCDs during HIV care and dyslipidemia after ART initiation as the main explanations for apparent differences in incidence after enrollment in care in people enrolled later in life (>50yo).

- As pointed out the lower prevalence in this study compared to other settings may be explained by lower screening? What do guidelines say about frequency of screening in PLHIV, is there data on gap between guidelines and real world (e.g. as done in https://www.ncbi.nlm.nih.gov/pubmed/28578613 for the Netherlands)? Could they also provide data on eg prevalence of hypertension in HIV-negative from both regions to support the hypothesis of genetic/risk factor does result in lower prevalence in Latin America? 

ANSWER: There is a lack of guidelines for prevention, screening, care and treatment for specific NCDs for the general population at the country level, not to mention guidelines for HIV care for people 50 years and older (page 19, lines 391-395, references #44 to #46 ). We agree that lower screening for non-communicable diseases could be a reason for lower estimates of prevalence in our study population. Nonetheless, there is little information on the gaps between recommendations/guidelines and practice, as we indicate in this new version with added sentences in this section. We address these issues in the Discussion of this revised version (Page 19, lines 391-395, references #40 to #42). 

- The discussion is somewhat short. Authors should elaborate on a number of points: 

o How do the results compare to other data from the region and beyond show eg the Brazil and Mexico data mentioned in intro

o What are the implications for clinical care (prevention, screening, guidelines, polypharmacy eg Smit et al Lancet ID 2015)

o What gaps remain (e.g. more information on proportion of those screened who are treated and controlled) and what data would need to be collected. This is a great opportunity to highlight not just a huge burden but also call for more research – use it �

o How generalisable are the results given that it includes only 5 countries from the region. E.g. how different are some regions in Latin America. 

o How much data do Mexico and Brazil contribute to CCASAnet data (not the main contributors when looking at numbers overall but not sure for >50yo). As the previous study into NCD from the region included data from Mexico and Brazil it would be good for authors to clearly highlighted added benefits of this study in discussion in relation to existing evidence (multi-cohort, larger number, more generalisable to whole region etc). 

o Would it be useful to have cohorts with controls as in US and Europe to better understand the excess risk of HIV disease and ART on NCDs

ANSWER: We appreciate this comment and the reviewer’s suggestions. We did 

substantial changes to Discussion section to address these points. Please see specifically in the manuscript: pages 17-18, lines 352-364, pages 19, lines 378-381, page 19, lines 384-388, page 19, lines 391-395, pages 19-20, lines 400-402; and page 18, lines 356-360.

General comments

- Authors could use epidemiological terminology e.g prevalence instead of frequency, proportion etc

ANSWER: Abstract- Background: lines 42-43, we changed frequency to prevalence and incidence.

Introduction: We limited the use of the term frequency. We use it when referring to both measures (prevalence and incidence), and use the specific terms when appropriate (see, for instance, lines 92-93 in page 4, last paragraph of Introduction)

Methods: We extended this section and added paragraphs to provide a more detailed description of our analysis. In this section, we specifically define how we use these different measures of frequency. We now explicitly describe how we used estimated prevalence and incidence in the study.

Results: We changed the term Proportion for Prevalence in the title and legend of Figure 2. We changed the term Frequency for Prevalence In the title and legend of Table 2.

- When reporting on which NCDs contribute the largest burden, authors could focus on reporting the 3 main NCDs. The number of ‘main’ NCDs reported varies between abstract, results and discussion. 

ANSWER: We thank the reviewer for pointing these inconsistencies. We reviewed all the manuscript and modified the text in the Abstract and Results sections to make sure the information is consistent along the manuscript. 

- For figures authors may consider using increasingly darker shades from 0 to multimorbidity instead of a darker colour in the middle for 1NCD? If staying in grey shades and increasing in darkness makes it tricky to read consider using patterns (e.g. no pattern=0 NCDs, light pattern is 1 and dark pattern is 2+)

ANSWER: We followed the reviewer´s advice and modify figures as suggested. In Figure 1, we also added a table documenting the number of patients >50yo receiving care each year.

Minor comments

- In abstract could authors be more explicit in abstract that NCD prevalence was estimated amongst PLHIV (not general population attending centres).

ANSWER: We added wording in the abstract to make clear that we studied only people receiving care for HIV and not the general population. This sentence in the Background of the abstract now is read as follows: “We estimated the prevalence and incidence of non-communicable diseases (NCD), including multimorbidity, among people 50 years of age or older (≥50yo) receiving HIV care during 2000-2015 in six centers affiliated with the Caribbean, Central and South American network for HIV epidemiology (CCASAnet).” (Please see page 2, lines 42-45)

- In conclusions of abstract authors may also like to elude to the fact that the burden seems to be increasing over time

ANSWER: We appreciate this comment. In the new version of the manuscript we stated: “The prevalence of NCDs and multimorbidity in people ≥50 years receiving care for HIV in CCASAnet centers in Latin America increased substantially in the last 15 years.” (Please see, pages 2-3, lines 58-60)

- In introduction could authors state that the study was “from Name et al” instead of “from our colleagues” colleagues?

ANSWER: We changed this section of the main text as suggested by the reviewer. The new version (Page 3, Lines 75-78) says: “A recent study by Carraquiry et al in our cohort suggested that those starting cART at 50 years or older had an increased risk of death independent of CD4 count or AIDS when compared with people starting cART at younger ages, despite decreased risk of virological failure and treatment modification [9]”.

- In introduction when reporting on reasons for excess mortality authors should say “possible reason could be the increased presence” instead of “is”

ANSWER: We changed the text as suggested.

- Discussion could benefit from use of paragraph to distinguish individual sections. 

ANSWER: We added subheadings to paragraphs in the Discussion section to help distinguish individual sections and better organize this section.

- In discussion authors state that HTN, dyslipidaemia, psychiatric disorders AND diabetes are the main NCDs but diabetes was not mentioned in result or abstract. 

ANSWER: We reviewed and modified the text in the Abstract and Results sections to make sure the information is consistent along the manuscript. 

- It would strengthen discussion to mention what drives observed trends when discussion difference in NCD burden at enrolment and end of follow-up (higher incidence)

ANSWER: We discussed this, specifically in the second paragraph of the Discussion section (Page 16, lines 326-331), that says: “We observed a lower prevalence of NCDs at the start of follow-up in patients ≥50yo at enrollment, but at the end of follow-up the prevalence of any comorbidity and multi-morbidity was similar regardless of age at enrollment. While people enrolled in HIV care at 50yo or older had an apparently lower prevalence of NCDs at baseline and higher incidence during follow-up, increased opportunities for diagnosis after enrollment in regular medical care for HIV-infection are likely to account for most of this difference”. 

We also added in a brief statement, about the potential effect of ART over serum lipids, and how this might have contributed to our findings (Pages 16-17 , lines 331-336 ) “Antiretroviral initiation-associated dyslipidemia might also partially explain these differences, as almost all patients in our centers were initiated on efavirenz- or boosted lopinavir-based regimens with ziduvudine and lamivudine during the study period [20, 22]; and dyslipidemia, a common complication of former first-line ART regimes [30], was the most commonly registered incident NCD among patients ≥50yo at enrollment”

- I am not sure I understand the sentence “Another potential limitation is that the prevalence …”

- 

ANSWER: We were trying to acknowledge that prevalence of NCDs and multimorbidity are parameters that provide general information about health and need of services for this population, but not a precise indicator of specific healthcare needs in this group. Comprehensive healthcare for aging adults takes into consideration other variables that might be more relevant such as disability, polypharmacy, neurocognitive decline and other geriatric syndromes. We modified the text accordingly, and hope this change contributes to clarify this point (Please see, page 19, lines 378-381). 

- The sentence “As such, we focus… “is misleading given the point on underdiagnosis. Authors may want instead to comment on gaps could be addressed for future analysis (analysis of screening practice to see if it is true underdiagnosis, that underscreening compared to guidelines has also been reported in HIC eg https://www.ncbi.nlm.nih.gov/pubmed/28578613. 

ANSWER: We agree and modified this part of the Discussion section as suggested by the reviewer (Page 19, lines 384-391). In the new version, we deleted this sentence and included this: “Despite systematic efforts to improve data quality and management in CCASAnet [25,26], this data is subject to potential information biases: NCDs diagnoses were included solely if recorded in the medical record by the provider, and clinical file management practices may have differed by site and over time; and screening for NCDs is not standardized across center, as previously described [39]. Health care seeking behavior, also affects recognition and capture of diagnoses for these conditions and this was one of the reasons we analysed separately groups by age at enrollment in care. Thus, underdiagnosis and under-reporting of NCDs might have led us to underestimate the real frequency of these disorders [40-42].”

Journal Requirements

ANSWER: Our manuscript meets the style requirements. 

1. Thank you for including your ethics statement: "Institutional review board approval was obtained locally for each participating site and for the DCC at Vanderbilt. In each of the sites contributing data to this study, ethical regulations and policies permit retrospective analysis of de-identified clinical data without informed consent when research is approved by an Institutional Review Board or appropriately constituted ethics committee. "

ANSWER: Please see the ethics statement in the methods section which includes the names of each ethics committee. 

“Institutional review board approval was obtained locally for each participating site and for the DCC at Vanderbilt. In each of the sites contributing data to this study, ethical regulations and policies permit retrospective analysis of de-identified clinical data without informed consent when research is approved by an Institutional Review Board or appropriately constituted ethics committee. Argentina – Comité de Bioética de Fundación Huésped. Brazil- Comité de Ética em Pesquisa (CEP), Fundacao Oswaldo Cruz, Instituto Nacional de Infectologia Evandro Chagas. Comissao Nacional de Etica em Pesquisa (CONEP). Chile- Comite Etico Científico del Servicio de Salud Metropolitano Central. Haiti- Comité des Droits Humains des Centres GHESKIO, and Weill Cornel Medical College Institutional Review Board .Honduras –Comité de Ética en Investigación Biomédica de la Unidad de Investigación Científica de la Universidad Nacional autónoma de Honduras. México – Comité de Ética en Investigación del Instituto Nacional de Ciencias Médicas y Nutrición Salvador Zubirán. Perú– Comité Institucional de Ética, Universidad Peruana Cayetano Heredia. VUMC – Vanderbilt University Institutional Review Board.”

2. Thank you for including your competing interests statement; "Pablo F Belaunzaran-Zamudio, Yanink Caro-Vega, Mark J Giganti, Brenda Crabtree-Ramírez, Bryan E Shepherd, Carina Cesar, Rodrigo C Moreira, Fernando Mejia, Marcelo Wolff, Jean W. Pape, Denis Padgett and Catherine C McGowan have no conflicts to declare. Juan Sierra-Madero reports personal fees and non-financial support from Gilead, non-financial support from MSD, grants from BMS, grants from Pfizer, and personal fees from Jansen, all outside the submitted work. "

ANSWER: The competing interests statement below includes now some sentences explaining the sharing data principles for CCASAnet as you suggested: 

"Pablo F Belaunzaran-Zamudio, Yanink Caro-Vega, Mark J Giganti, Brenda Crabtree-Ramírez, Bryan E Shepherd, Carina Cesar, Rodrigo C Moreira, Fernando Mejia, Marcelo Wolff, Jean W. Pape, Denis Padgett and Catherine C McGowan have no conflicts to declare. Juan Sierra-Madero reports personal fees and non-financial support from Gilead, non-financial support from MSD, grants from BMS, grants from Pfizer, and personal fees from Jansen, all outside the submitted work. This does not alter our adherence to PLOS ONE policies on sharing data and materials. Complete data for this study cannot be publicly shared because of legal and ethical restrictions. The Principles of Collaboration under which the CCASAnet multi-national collaboration was founded and the regulatory requirements of the different countries' IRBs require the submission and approval of a project concept sheet by the CCASAnet Executive Committee and the principal investigators at participating sites. All datasets provided by CCASAnet are de-identified according to HIPAA Safe Harbor guidelines. Since re-identification of de-identified datasets may be possible when they are combined with publicly available datasets, CCASAnet promotes the signing of a Data Use Agreement before HIV clinical data can be released. Instructions for how to obtain CCASAnet data are outlined on the CCASAnet website: https://www.ccasanet.org/collaborate/. "

Best Regards, 

Yanink Caro-Vega

Unidad del Paciente Ambulatorio (UPA), 5to piso

Vasco de Quiroga # 15

Col. Sección XVI

Delegación Tlalpan; C.P. 14000

México D.F.; México

Tel. +52 (55)54870900; ext, 5505 

Fax. +52 (55)55130010

yanink.caro@infecto.mx

---

## [Decision Letter · Decision Letter 1]

25 Mar 2020

PONE-D-19-27737R1

Frequency of non-communicable diseases in people 50 years of age and older receiving HIV care in Latin America.

PLOS ONE

Dear PhD Caro-Vega,

Thank you for submitting your manuscript to PLOS ONE. After careful consideration, we feel that it has merit but does not fully meet PLOS ONE’s publication criteria as it currently stands. Therefore, we invite you to submit a revised version of the manuscript that addresses the points raised during the review process.

An additional reviewer made important remarks in the manuscript. The comments are appended below. We would appreciate receiving your revised manuscript by May 09 2020 11:59PM. To enhance the reproducibility of your results, we recommend that if applicable you deposit your laboratory protocols in protocols.io, where a protocol can be assigned its own identifier (DOI) such that it can be cited independently in the future. For instructions see: http://journals.plos.org/plosone/s/submission-guidelines#loc-laboratory-protocols

We look forward to receiving your revised manuscript.

Kind regards,

Bruno Pereira Nunes, Ph.D.

Academic Editor

PLOS ONE

Reviewers' comments:

Reviewer's Responses to Questions

**Comments to the Author**

1. If the authors have adequately addressed your comments raised in a previous round of review and you feel that this manuscript is now acceptable for publication, you may indicate that here to bypass the “Comments to the Author” section, enter your conflict of interest statement in the “Confidential to Editor” section, and submit your "Accept" recommendation.

Reviewer #3: (No Response)

2. Is the manuscript technically sound, and do the data support the conclusions?

Reviewer #3: Partly

3. Has the statistical analysis been performed appropriately and rigorously? 

Reviewer #3: No

4. Have the authors made all data underlying the findings in their manuscript fully available?

Reviewer #3: No

5. Is the manuscript presented in an intelligible fashion and written in standard English?

Reviewer #3: No

6. Review Comments to the Author

Reviewer #3: The report of the Frequency of non-communicable diseases in people 50 years of age and older receiving HIV care in Latin America by Belaunzaran-Zamudio and colleagues presents information about the frequency of non-communicable diseases (NCDs) in persons receiving treatment for HIV related conditions; an area where there is limited data in Latin America and the Caribbean.

However, in my opinion, there are several elements of this paper that still require justification and/or further clarification.

The authors note the high prevalences of NCDs in single center cohorts of older persons being treated for HIV in Mexico and Brazil (lines 83-86), but do not describe whether these rates differ from comparable rates in groups without HIV.

A major concern of this paper is the methodology used in the analysis as the authors do not provide satisfactory justification for dichotomizing the study sample. They state; ‘and in our own exploratory data analysis, people enrolled in care at older age appear to be fundamentally different than people that enrolled younger and aged receiving care and ART’ [lines 164 – 166]. As such the authors should present supporting data to support this view and to justify the methods that they used. They cite two references by Guardali et al (references 11 and 18). When one evaluates the BMC Geriatrics reference (11), the factor which is associated with NCD risk in persons with HIV is not age, but rather duration of HIV treatment.

This study is presents as both a prevalence study and a cohort study where incident outcomes are reported. A concern is the very short median period of follow up of only 3.7 years at ages ≥50 years [233]. Given that over the 15-year period of review, persons would have died, been lost to follow up or excluded for reasons such as non-attendance, in the absence of more comprehensive data regarding participation by study subjects, it is difficult to understand actual participation by study subjects or potential biases that might have resulted from non-participation/losses to follow up, or indeed patterns of participation.

Additionally, regarding reported prevalence data, there is the possibility of biases linked to differences in survival or participation for those without NCDs compared to those with NCDs. The authors note that ‘We observed a lower prevalence of NCDs at the start of follow-up in patients ≥50yo at enrollment, [line 327]. Advancing this argument, then the reported increased prevalence of NCDs over time, might have been a feature of better access to improved care associated with increased survival, but greater multimorbidity. This is relevant as participating subjects repeatedly contribute to annual prevalence data.

The observation regarding the lower occurrence of NCDs in this cohort of persons with HIV in Latin America compared to higher income countries [lines 342-346] clearly is the result of several factors which might include differences in participation in care, quality of care, study methodological issues (sampling, biases etc.. ). It would be important and relevant for the authors to evaluate whether the burden of NCDs is in fact different in this population of persons being treated for HIV compared to similar unaffected populations in the Region.

7. PLOS authors have the option to publish the peer review history of their article (what does this mean?). If published, this will include your full peer review and any attached files.

Reviewer #3: No

---

## [Author Response · Author response to Decision Letter 1]

11 May 2020

Response to Reviewers

Dear Dr. Pereira:

We appreciate the interest of PLoS ONE in our manuscript and welcome the feedback of a new reviewer. We think these comments helped us to improve our manuscript and contributed to convey our ideas clearer in the new draft we are submitting for your consideration. Next, we respond to each point raised by the new reviewer, addressing the changes each comment lead to in the new version. In addition to the specific changes listed below, we reorganized a subheading of the Discussion section (Limitations) to better organize the changes and for clarity purposes. All changes were tracked using the Words function.

Comments by Reviewer #3: 

The report of the Frequency of non-communicable diseases in people 50 years of age and older receiving HIV care in Latin America by Belaunzaran-Zamudio and colleagues presents information about the frequency of non-communicable diseases (NCDs) in persons receiving treatment for HIV related conditions; an area where there is limited data in Latin America and the Caribbean.

However, in my opinion, there are several elements of this paper that still require justification and/or further clarification.

Comment: The authors note the high prevalences of NCDs in single center cohorts of older persons being treated for HIV in Mexico and Brazil (lines 83-86), but do not describe whether these rates differ from comparable rates in groups without HIV.

Author´s response: In the revised manuscript, we have included some published data on NCDs among persons without HIV. We used published information from age-specific, population-based estimates from the Health and Nutrition National Survey in Mexico, the Telephonic Survey for Health Risk Surveillance in Brazil, a population-based study in rural, western Honduras, and a national urban population-based longitudinal study in Peru (PERUDIAB). These data provide some context for our analysis. In these surveys, the prevalence of a few NCDs in people of 50 years of age and older is actually high. We added the next sentences in the third paragraph (page 4, lines 81-88) of the Introduction section (and added the citations for new references used to support the statement):

“Among the general population of people aged 50 or older in Latin America, the prevalence of NCDs is high. For example, recent studies from Brazil [11,12], Honduras [13], Mexico [14-16], and Peru [17] have all reported estimates of the prevalence of diabetes mellitus (DM) in older populations to be 12% or higher. Among those between the ages of 60 and 70 in Mexico, the estimated prevalence was as high as 36% [14]. The percentage of people 50 years and older with hypertension (HTN) was estimated to be greater than 40% in urban and rural settings in Mexico [14-16], Brazil [11,12], and Honduras [13]. Dyslipidemia was also higher than 35% in both Brazil and Mexico [11,12,16].” 

There are of course some challenges to comparing our estimates with those from the general populations. In particular, the methodologies for collecting the information are different, ascertainment may vary, and there could be discrepancies/bias in who is included in the estimates. We also modified the Discussion section to elaborate on possible explanations for an overall lower prevalence of some of these NCDs than in the general population in our region. Thus, we added a paragraph in the subheading Results of the study in context (page 18, lines 380-384):

“We observed a lower prevalence of NCDs in our cohort relative to prevalence estimates corresponding to the general population in Latin America. [11-17]. It is unclear whether this is due to incomplete ascertainment of NCDs or selection biases that make a cohort of patients longitudinally receiving treatment and care at a health center not comparable with a representative sample of the general population.”

Comment: A major concern of this paper is the methodology used in the analysis as the authors do not provide satisfactory justification for dichotomizing the study sample. They state; ‘and in our own exploratory data analysis, people enrolled in care at older age appear to be fundamentally different than people that enrolled younger and aged receiving care and ART’ [lines 164 – 166]. As such the authors should present supporting data to support this view and to justify the methods that they used. They cite two references by Guardali et al (references 11 and 18). When one evaluates the BMC Geriatrics reference (11), the factor which is associated with NCD risk in persons with HIV is not age, but rather duration of HIV treatment.

Author´s response: We appreciate this comment by the reviewer, as it allowed us to better contextualize our work. In this study, we build-up from our previous research (Caro-Vega Y, et al. Epidemiol Infect. 2018) where we described how the percentage of patients over 50 years increased from 8% to 24% from 2000 to 2015. During this period, people enrolled younger in care and receiving ART contribute differently to the older HIV population growth than those enrolled at older ages. We decided to maintain this categorization to explore how much each group might have contributed to the prevalence of comorbidities and multi-morbidity in our region. To clarify this point, we modified the Methods section with the following sentences (page 7, lines 164-170, subheading Study population)

“While the classification of ‘older PLWHIV” is not explicitly defined, a threshold of 50 years of age is commonly used to indicate an aging or older HIV population [35]. In our own research, we have used this threshold to assess how people enrolled at younger ages and receiving ART have contributed to growth of the older PLWHIV population compared to those enrolled at older ages [7]. We maintain this categorization to explore how much each group might contribute to the prevalence of comorbidities and multi-morbidity in our region.” 

The results presented in the studies by Guaraldi and collaborators (Guaraldi et al, BMC Geriatrics 2018; Guaraldi et al. PLoS ONE 2015) shows that both, age and time since diagnosis of HIV (not exactly time on ART, which was not measured but which is likely positively correlated with time since diagnosis) are associated with some comorbidities, multimorbidity and polypharmacy, as can be inferred from results summarized in tables 1 & 2, and figures 2 & 4 of that manuscript. 

These results support our decision to categorize our study population in those that enrolled in care before versus after 50 years of age, as those diagnosed and enrolled in care at younger ages have been enrolled in care longer, exposed to ART longer, and probably have been HIV-infected longer than those enrolled in care at older ages. We provide this as a rationale for our decision (see page 4 & 5, lines 102-105).

“Patients were stratified by age at enrollment in care (<50 years at enrollment and ≥ 50 years at enrollment) given possible differences in time of ART exposure, previous health history, HIV duration and severity, clinical outcomes, and mortality between people aging in care and people diagnosed at older age [18, 26].”

Furthermore, the results presented in Table 1 support our view, and make evident that not only the mean age of these groups is quantitatively different but crucially, people enrolled in care at older ages have different time in follow-up than those enrolled younger and started receiving care more recently. Our results and conclusions are overall in agreement with those by Guaraldi in that manuscript. 

In response to the reviewers´ comments, we added some additional information relevant to this subject. We modified the text in the Methods section to better explain our rationale for this decision (page 7, lines 164-170). We also added 2 rows In Table 1 summarizing the information about the frequency of ART use and time on ART in each group. We mentioned these differences in the Results, page 11 (lines 240-243). This information strengthens our rationale for categorizing this population based in age at enrollment in care.

In the modified text in the Methods section, we now state the following:

“While the classification of ‘older PLWHIV” is not explicitly defined, a threshold of 50 years of age is commonly used to indicate an aging or older HIV population [35]. In our own research, we have used this threshold to assess how people enrolled at younger ages and receiving ART have contributed to growth of the older PLWHIV population compared to those enrolled at older ages [7]. We maintain this categorization to explore how much each group might contribute to the prevalence of comorbidities and multi-morbidity in our region.” 

In the modified text in the Results section, we now state the following:

“Patients <50yo at enrollment were followed a median time of 3.1 years after reaching 50 years (cumulative time of 7,821 years in follow-up). In contrast, those ≥50yo at enrollment were followed for 4.6 years (8,064 cumulative years in follow-up).”

We agree with the reviewer that this approach has limitations that need to be discussed in our manuscript. In the new draft, we discuss the limitations and advantages of our approach in the context of previously published literature (see pages 19-20, lines 415-420), where we state the following:

“We classified our study population in two groups based on their age at enrollment in care. We acknowledge this is an arbitrary distinction that does not necessarily accurately depict a complex aging biological process intertwined with the effects of long-term toxicity of ART. Nonetheless, ours is a descriptive study aiming to quantify this emerging problem and we found this a useful solution to describe separately subgroups in our population of interest that present different morbidity patterns.”

Comment: This study is presents as both a prevalence study and a cohort study where incident outcomes are reported. A concern is the very short median period of follow up of only 3.7 years at ages ≥50 years [233]. Given that over the 15-year period of review, persons would have died, been lost to follow up or excluded for reasons such as non-attendance, in the absence of more comprehensive data regarding participation by study subjects, it is difficult to understand actual participation by study subjects or potential biases that might have resulted from non-participation/losses to follow up, or indeed patterns of participation.

Author´s response: Patients <50yo at enrollment were followed for a cumulative time of 7,821 years; and those ≥50yo at enrollment were followed for 8,064 cumulative years We do not consider a median follow-up time of almost 4 years as short. 

While the study period encompasses 15 years of enrollment and follow-up, 50% of enrollment in this cohort occurred between 2010 and 2015 (see Table 1). Most patients (77%) were followed-up until 2015 and administratively censored (79% in patients enrolled<=50 and 73% in those enrolled ≥50yo), 15% were lost to follow-up (LTFU) in both groups, and 8% deaths (5% in patients enrolled<=50 and 11% in those enrolled ≥50yo). We included this information in the Results section of new draft of the manuscript (see Page 11, lines 246-253). 

“While the study period encompasses a 15-year period (2000-2015), 50% of all person time of observation in this study occurred between 2010 and 2015 (Table 1). Most patients were followed-up until 2015 and were administratively censored (79% in patients enrolled<=50 and 73% in those enrolled ≥50yo). The remaining participants were lost to follow-up (17%) or died (8%). Compared to those ≥50yo at enrollment, a higher proportion of patients <50yo at enrollment were administratively censored (79% vs 73%, respectively) and a lower proportion died (5% vs 11%, respectively).”

Comment: Additionally, regarding reported prevalence data, there is the possibility of biases linked to differences in survival or participation for those without NCDs compared to those with NCDs. The authors note that ‘We observed a lower prevalence of NCDs at the start of follow-up in patients ≥50yo at enrollment, [line 327]. Advancing this argument, then the reported increased prevalence of NCDs over time, might have been a feature of better access to improved care associated with increased survival, but greater multimorbidity. This is relevant as participating subjects repeatedly contribute to annual prevalence data.

Author´s response: We acknowledge this observation is likely the result of better access to improved care at different time in for each age-group. Please see discussion section, at “Frequency of NCDs” subsection, page 16, lines 344-347: 

“While people enrolled in HIV care at 50yo or older had an apparently lower prevalence of NCDs at baseline and higher incidence during follow-up, increased opportunities for diagnosis after enrollment in regular medical care for HIV-infection are likely to account for most of this difference.”

Comment: The observation regarding the lower occurrence of NCDs in this cohort of persons with HIV in Latin America compared to higher income countries [lines 342-346] clearly is the result of several factors which might include differences in participation in care, quality of care, study methodological issues (sampling, biases etc.. ). It would be important and relevant for the authors to evaluate whether the burden of NCDs is in fact different in this population of persons being treated for HIV compared to similar unaffected populations in the Region.

 Author´s response: The purpose of this study is to quantify the frequency of NCDs in this population, which to date has been understudied in our region. We added text regarding the frequency of NCDs among older people in the general population in Latin America, to provide more background about the burden of these disorders in the general population in Latin America.

Please see introduction, page 4, lines 81-88: 

“Among the general population of people aged 50 or older in Latin America, the prevalence of NCDs is high. For example, recent studies from Brazil [11,12], Honduras [13], Mexico [14-16], and Peru [17] have all reported estimates of the prevalence of diabetes mellitus (DM) in older populations to be 12% or higher. Among those between the ages of 60 and 70 in Mexico, the estimated prevalence was as high as 36% [14]. The percentage of people 50 years and older with hypertension (HTN) was estimated to be greater than 40% in urban and rural settings in Mexico [14-16], Brazil [11,12], and Honduras [13]. Dyslipidemia was also higher than 35% in both Brazil and Mexico [11,12,16].” 

We also elaborate in the discussion section, please see page 18 lines 380-384: 

“We observed a lower prevalence of NCDs in our cohort relative to prevalence estimates corresponding to the general population in Latin America. [11-17]. It is unclear whether this is due to incomplete ascertainment of NCDs or selection biases that make a cohort of patients longitudinally receiving treatment and care at a health center not comparable with a representative sample of the general population.”

We also acknowledge that differences with previous studies in high-income countries are likely to be multifactorial, including characteristics of the structure of the cohort (please see pages 17-18, lines 360-371): 

“Differences in the frequency of environmental, genetic and individual risk factors for NCDs might explain these disparities [41,42].Previous studies in Italy found that people aging while receiving HIV care more frequently had several specific NCDs, multimorbidity, and polypharmacy than people of similar age diagnosed and enrolled in care later in life [18,26]. In contrast, we saw that patients enrolled in care after 50 years of age had a lower prevalence of NCDs than those who reached 50 years of age while in care, whereas those enrolled after 50 years had a higher incidence of NCDs. This discrepancy may potentially be explained by the shorter time in observation for both groups in our study and considerable larger differences of time living with HIV between groups in Italy (≥10 years in comparison with around 5 years in this study), rather than age itself [18,22,26].”

---

## [Editor Report · Decision Letter 2]

18 May 2020

Frequency of non-communicable diseases in people 50 years of age and older receiving HIV care in Latin America.

PONE-D-19-27737R2

Dear Dr. Caro-Vega,

We are pleased to inform you that your manuscript has been judged scientifically suitable for publication and will be formally accepted for publication once it complies with all outstanding technical requirements.

With kind regards,

Bruno Pereira Nunes, Ph.D.

Academic Editor

PLOS ONE
---

## [Editor Report · Acceptance letter]

22 May 2020

PONE-D-19-27737R2 

Frequency of non-communicable diseases in people 50 years of age and older receiving HIV care in Latin America. 

Dear Dr. Caro-Vega:

I am pleased to inform you that your manuscript has been deemed suitable for publication in PLOS ONE. Congratulations! Your manuscript is now with our production department. 

With kind regards,

on behalf of

Dr. Bruno Pereira Nunes 

Academic Editor

PLOS ONE